# MULTIMODALQA: COMPLEX QUESTION ANSWERING OVER TEXT, TABLES AND IMAGES

**Alon Talmor**[*,1,2]   **Ori Yoran**[*,1,2]   **Amnon Catav**[*,2]   **Dan Lahav**[*,2]   **Yizhong Wang**[3]
**Akari Asai**[3]   **Gabriel Ilharco**[3]   **Hannaneh Hajishirzi**[2,3]   **Jonathan Berant**[1,2]
[1]The Allen Institute for AI,  [2]Tel-Aviv University,  [3]University of Washington
{alont,oriy,jonathan}@allenai.org
{amnoncatav,lahav}@mail.tau.ac.il
{yizhongw,akari,gamaga,hannaneh}@cs.washington.edu

## ABSTRACT

When answering complex questions, people can seamlessly combine information from visual, textual and tabular sources. While interest in models that reason over multiple pieces of evidence has surged in recent years, there has been relatively little work on question answering models that reason *across* multiple modalities. In this paper, we present MULTIMODALQA (MMQA): a challenging question answering dataset that requires joint reasoning over text, tables and images. We create MMQA using a new framework for generating complex multi-modal questions at scale, harvesting tables from Wikipedia, and attaching images and text paragraphs using entities that appear in each table. We then define a formal language that allows us to take questions that can be answered from a single modality, and combine them to generate *cross-modal* questions. Last, crowdsourcing workers take these automatically generated questions and rephrase them into more fluent language. We create 29,918 questions through this procedure, and empirically demonstrate the necessity of a multi-modal multi-hop approach to solve our task: our multi-hop model, *ImplicitDecomp*, achieves an average $F_1$ of 51.7 over cross-modal questions, substantially outperforming a strong baseline that achieves 38.2 $F_1$, but still lags significantly behind human performance, which is at 90.1 $F_1$.

## 1 INTRODUCTION

When presented with complex questions, people often do not know in advance what source(s) of information are relevant for answering it. In general scenarios, these sources can encompass multiple modalities, be it paragraphs of text, structured tables, images or combinations of those. For instance, a user might ponder *"When was the famous painting with two touching fingers completed?"*, if she cannot remember the exact name of the painting. Answering this question is made possible by integrating information across both the textual and visual modalities.

Recently, there has been substantial interest in question answering (QA) models that reason over multiple pieces of evidence (*multi-hop* questions (Yang et al., 2018; Talmor & Berant, 2018; Welbl et al., 2017)). In most prior work, the question is phrased in natural language and the answer is in a context, which may be a paragraph (Rajpurkar, 2016), a table (Pasupat & Liang, 2015), or an image (Antol et al., 2015). However, there has been relatively little work on answering questions that require integrating information *across* modalities. Hannan et al. (2020) created MANYMODALQA: a dataset where the context for each question includes information from multiple modalities. However, the answer to each question can be derived from a *single modality* only, and no *cross-modality reasoning* is needed. Thus, the task is focused on identifying the relevant modality. Recently, Chen et al. (2020b) presented HYBRIDQA, a dataset that requires reasoning over tabular and textual data. While HYBRIDQA requires cross-modal reasoning, it does not require visual inference, limiting the types of questions that can be represented (See Table 1 for a comparison between the datasets).

---

[*]   The authors contributed equally

Figure 1: Example of a *MMQA* question, answer and context. In green are the text modality question and answer, and in red the image modality. The table is used to perform the year comparison between the answers of the text and image question parts.

In this work, we present *MMQA*, the first large-scale (29,918 examples) QA dataset that requires integrating information across free text, semi-structured tables, and images, where 35.7% of the questions require cross-modality reasoning. Figure 1 shows an example question: *"Which B.Piazza title came earlier: the movie S. Stallon's son starred in, or the movie with half of a lady's face on the poster?"*. Answering this question entails (i) decomposing the question into a sequence of simpler questions, (ii) determining the modalities for the simpler questions and answering them, i.e., information on the poster is in an image, the information on *"S. Stallon's son"* is in free text, and the years of the movies are in the table, (iii) combining the information from the simpler questions to compute the answer: *"Tell Me that you love me, Junie Moon"*.

Our methodology for creating *MMQA* involves three high-level steps. (a) *Context construction*: we harvest tables from Wikipedia, and connect each table to images and paragraphs that appear in existing Reading Comprehension (RC) datasets (Kwiatkowski et al., 2019; Clark et al., 2019; Yang et al., 2018); (b) *Question generation*: Following past work (Talmor & Berant, 2018), we use the linked structure of the context to automatically generate questions that require multiple reasoning operations (composition, conjunction, comparison) across modalities in pseudo-language ; (c) *Paraphrasing*: we use crowdsourcing workers to paraphrase the pseudo-language questions into more fluent English.

To tackle *MMQA*, we introduce *ImplicitDecomp*, a new model that predicts a program that specifies the required reasoning steps over different modalities, and executes the program with dedicated text, table, and image models. *ImplicitDecomp* performs multi-hop multimodal reasoning without the need for an explicit decomposition of the question.

| Dataset | Size | Full-wiki | Uses images | Multi-hop |
|---------|------|-----------|-------------|-----------|
| MANYMODALQA | 10K | ✗ | ✓ | ✗ |
| HYBRIDQA | 70K | ✗ | ✗ | ✓ |
| MULTIMODALQA | 30K | ✓ | ✓ | ✓ |

Table 1: A comparison of MULTIMODALQA to MANYMODALQA and HYBRIDQA. We compare dataset size, use of images, and whether the dataset supports multi-hop questions and an open-domain full-wiki setup.

We empirically evaluate *MMQA* by comparing *ImplicitDecomp* to strong baselines that do not perform cross-modal reasoning and to human performance. We find that on multimodal questions, *ImplicitDecomp* improves $F_1$ from $38.2 \rightarrow 51.7$ over a single-hop approach. Humans are able to reach 90.1 $F_1$, significantly outperforming our best model. Because automatic evaluation is non-trivial, we also manually analyze human performance and find humans correctly answer 94.5% of the questions in *MMQA*. Finally, our dataset can be used in an open-domain setup over all of Wikipedia. In this setup, the $F_1$ of humans is 84.8.

To summarize, our key contributions are:

- *MMQA*: a dataset with 29,918 questions and answers, 35.7% of which require cross-modal reasoning.
- A methodology for generating multimodal questions over text, tables and images at scale.
- *ImplicitDecomp*, A model for implicitly decomposing multimodal questions, which improves on a single-hop model by 13.5 absolute $F_1$ points on questions requiring cross-modal reasoning.
- Our dataset and code are available at `https://allenai.github.io/multimodalqa`.

## 2 DATASET GENERATION

Our goal is to develop a method that allows generating complex questions over multiple modalities at scale. An overview of the methodology is captured in Figure 2. We first select a Wikipedia table as an anchor, to which we add images and texts paragraphs and obtain a *context*. Single modality questions are generated based on these contexts, and used to automatically create multimodal, multi-hop questions. AMT workers rephrase the questions into natural language, and finally distractor paragraphs and images are selected for each question. We now elaborate on the 6 steps of the process.

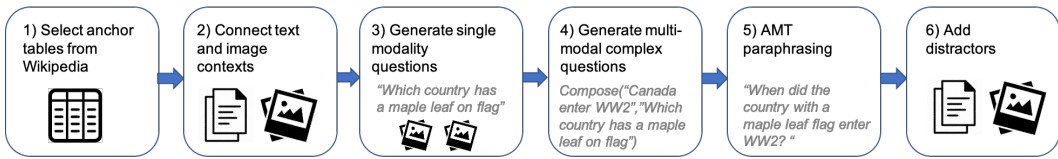

Figure 2: An overview of *MMQA* dataset generation process.

**2.1 Wikipedia tables as anchors** The 01-01-2020 English Wikipedia dump contains roughly $3M$ tables. We extracted all tables and selected those that meet the following criteria: (a) The tables contain 10-25 rows (b) At least 3 images are associated with the table. This results in a total of 700k tables. (see supp. material for more information). These tables are the anchors of our contexts, which we enrich with images and text for multimodal question generation. A key element of the tables are Wikipedia Entities (*WikiEntities*) that appear in them, i.e., concepts linked to other Wikipedia entries. We use them to connect different modalities, bridge questions, and solve ambiguities (details below).

**2.2 Connecting Images and Text to Tables** *Images.* We consider two cases: (a) in-table images and (b) images from pages of linked *WikiEntities*. In the former, the images are featured inside the table cells. In the latter, the table contains a column of *WikiEntities* that potentially have images, e.g. a table describing the filmography of an actor often contains a column of film names, which may have posters in their respective pages. To associate entities with their representative image, we map entities and their profile images in their Wikipedia pages. Overall, we obtain 57,713 images, with 889 in-table images and 56,824 *WikiEntities* images. *Text.* We build on texts from contexts appearing in existing reading comprehension datasets. We elaborate on this process next.

**2.3 Generating Single-Modality Questions** *Tables.* We generate pseudo-language table questions in the following form *"In [table title] of [Wikipedia page title] which cells in [column X] have the [value Y] in [column Z]?"*. We additionally support numeric computations over columns classified as dates or numbers, such as min and max values, e.g., *"In [Doubles] of [WCT Tournament of Champions], what was the MOST RECENT [Year](s) where the [Location] was [Forest Hills]"*.

*Images.* We use crowdsourcing to generate single-modality questions about images. We generated two types of image questions, based on the images we retrieved from the previous step: (i) questions over a single image, (ii) questions over a list of images.

When generating single-image questions, we show Amazon Mechanical Turk (AMT) crowd workers an image alongside its *WikiEntity*, and ask them to phrase a question about the image with the entity being the focus of the question. E.g, if the entity is *"Roger Federer"*, a potential question is *"What's the hair color of Roger Federer?"*. For questions to have meaning in an open-domain setting, we primed AMT workers to ask questions that correspond to "stable" features, i.e., features that are unlikely to change in different images and are thus appropriate in an open-domain setting.

| Type | Q&A | % |
|---|---|---|
| TextQ | *What was the territorial capital of the territory opposing Ohio in the Toledo War? Detroit* | 31.0 |
| TableQ | *Does the German state Baden-Wurttemberg or Thuringia have more residents? Baden-Württemberg* | 18.3 |
| ImageQ | *What weapon is the statue in Nottingham holding? bow* | 8.9 |
| Compose(TextQ,TableQ) | *At what age did the Cleveland Cavaliers player with 6190 rebounds enter the NBA? 19* | 7.8 |
| ImageListQ | *What is the common name of the bush warbler in Thailand that has an orange stripe above its eye? Chestnut-crowned bush warbler* | 6.1 |
| Compose(TableQ,ImageListQ) | *The film that starred Chris Ellison where a man was holding a newspaper on the poster, was released what year? 1988* | 5.4 |
| Compose(ImageQ,TableQ) | *On the poster for the TV show in which Tom Mison played Dorian Crane, what kind of structure can be seen behind the two men? castle* | 4.5 |
| Compare(Compose(TableQ,ImageQ),TableQ) | *Which manufacturer has fewer wins at the First Data 500: Buick or the brand with a cross for a logo? Buick* | 3.5 |
| Compose(TableQ,TextQ) | *On what date did the original artist who sang Sweet Child of Mine have a concert at US Bank Stadium? July 30, 2017* | 3.2 |
| Intersect(TableQ,TextQ) | *Who was the artist for Damon Fox in 2006 who also sings "You got the moves like Jagger"? Christina Aguilera* | 2.6 |
| Compose(TextQ,ImageListQ) | *On the poster for the movie based on the book "Act like a Lady, Think Like a Man," how many people are there in total? nine* | 2.4 |
| Intersect(ImageListQ,TableQ) | *What covers of the Chandler Canterbury films from 2009 has more than one person? Powder Blue Balls Out, Gary the Tennis Coach, After.Life* | 2.3 |
| Compare(TableQ,Compose(TableQ,TextQ)) | *Did Chelsea or club that sings You'll Never Walk Alone rank higher in Deloitte Football Money League 2007? Chelsea* | 2.1 |
| Compose(ImageQ,TextQ) | *Did Gary Oldman take part in the movie whose poster features two men holding handguns, and which had Mark L. Smith as a writer? no* | 1.0 |
| Compare(Compose(TableQ,ImageQ), Compose(TableQ,TextQ)) | *Was the film that features a giant eye on its poster or the first Wolverine movie the earlier film that Scott Silver worked on? Requiem for a Dream* | 0.8 |
| Intersect(ImageListQ,TextQ) | *What common law state with an eagle on the flag has an institution in the North region of Division II of the NCCAA? Iowa* | 0.2 |

Table 2: All 16 compositional templates in *MMQA* with an example and their relative frequency.

For questions with a list of images, we use images that appear in the same column of a table. To generate these questions, AMT workers were given the images and asked to phrase a binary question about a distinctive feature of the entities that a subset of the images share. E.g., given a list of statues, the worker could ask *"Which of the statues features a horse?"* This process results in 2,764 single image questions and 7,773 list image questions that are later used to create multimodal questions.

***Text.*** To obtain questions answerable over text paragraphs we build on existing reading comprehension datasets: *Natural Questions (NQ)* (Kwiatkowski et al., 2019) consists of about $300K$ questions issued to the Google search engine. This dataset mostly contains simple questions where a single paragraph suffices to answer each question. *BoolQ* (Clark et al., 2019) contains $15,942$ yes/no questions, gathered using the same pipeline as NQ. *HotpotQA* (Yang et al., 2018) contains $112K$ training questions, where crowd workers were shown pairs of related Wikipedia paragraphs and were asked to author questions that require multi-hop reasoning over the paragraphs.

To use questions from the above datasets as building blocks for multi-hop multimodal questions, we unified them into a corpus that consists of triples of (i) a text question, (ii) an answer and (iii) 1-2 gold paragraphs from Wikipedia. We link a question to a table, by matching *WikiEntities* in the table to entities in the text of the question (see supplementary material for further details). Overall, we retrieved 6,644 questions from NQ, 1,246 from BoolQ and 4,733 from HotpotQA.

**2.4 Generating multimodal complex questions** We present an automatic method for creating at scale multimodal compositional questions (i,e., questions that require answering a sequence of sub-questions to conclude the final answer). Our first step is to introduce a formal language that allows to combine questions answerable from a single modality. Below we introduce the logical operations that allow to generate such pseudo-language (PL) questions, while keeping a formal representation of how they were constructed. In Table 2, we illustrate this process with all 16 different compositional templates used for question generation. We now describe our logical operations.

**Logical Operations** Functions in our formal language take arguments and return a PL partial question, as well as answers that can be a list of one or more strings, or a list of one or more *WikiEntities*. All operations have access to the full context. In addition, we prepend a prefix containing the Wikipedia table name and page title—e.g. *"In the Filmography of Brad Pitt,"*—to all our PL questions to support an open-domain QA setup. Our set of logical operations are:

1. **TABLEQ**: Returns a question from the table questions generated in §2.3, as well as a list of *WikiEntities* or a list of strings as answers.
2. **TEXTQ**: Returns a text corpus question (see §2.3) and a list of *WikiEntities* or strings as answers.
3. **IMAGEQ**: Returns a question about a single image associated with a *WikiEntity* and a single token answer from a fixed vocabulary (see §2.3).
4. **IMAGELISTQ**: Returns a question about a list of images and a list of *WikiEntities* corresponding to the images that answer the question (see §2.3).
5. **COMPOSE**$(\cdot, \cdot)$: Takes a PL question containing *a single WikiEntity* as a first argument, and a PL question that produces that *WikiEntity* as the output answer as its second argument. E.g., COMPOSE(*"Where was Barack Obama born?"*,*"Who was the 44th president of the USA?"*). The function replaces the *WikiEntity* in the first-argument PL question with the second-argument PL question and returns the resulting PL question (*"Where was the 44th president of the USA born?"*).
6. **INTERSECT**$(\cdot, \cdot)$: Takes two PL questions that return lists of more than one *WikiEntity*, and returns their intersection as the answer. The resulting PL question is of the form *"PL$_1$ and PL$_2$"* omitting PL$_2$'s first word (*"Who was born in Hawaii and is the parent of Sasha Obama?"*).
7. **COMPARE**$(\cdot, \cdot)$: Takes two PL questions each returning one *WikiEntity* that can be linked to one cell in the table, denoted by Ans$_1$, Ans$_2$. We first choose a numeric or date column in the table, if such exists. We then compare the values of this column corresponding to the rows of Ans$_1$ and Ans$_2$. Depending on the comparison outcome, output one of (Ans$_1$, Ans$_2$) as the operation answer. The PL question created is of the form *"What has compare-op numeric-column-name, PL$_1$ or PL$_2$?"* omitting PL$_1$ and PL$_2$'s first word. E.g. *"What has most recent creation year, the rocket of Appolo program, or the rocket of Gemini program?"*

**2.5 Paraphrasing using AMT** We used English-speaking AMT workers to paraphrase automatically-generated PL questions into natural language (NL). Each question was paraphrased by 1 worker and validated by 1-3 other workers. To avoid annotator bias (Geva et al., 2019), the number of annotators who worked on both the training and evaluation set was kept to a minimum. We also deployed a feedback mechanism, where workers receive a bonus if a baseline model correctly answered the question after their first paraphrasing attempt, but incorrectly after they refined the paraphrase. See supp. material for print-screens of the AMT annotator interface.

To generate diversity, workers got a bonus if the normalized edit distance of a paraphrase compared to the PL question was higher than 0.7. A total of 971 workers were involved, and 29,918 examples were produced with an average cost of 0.33$ per question. We split the dataset into 23,817 training, 2,441 development (dev.), and 3,660 test set examples. Context components in the dev. and test sets are disjoint, and were constructed from a disjoint set of single-modality questions.

A shortcoming of our method for automatically generating examples is that the question distribution does not come from a "natural" source. We argue that developing models that are capable of performing reasoning over multiple modalities is an important direction and *MMQA* provides an opportunity to develop and evaluate such models. Moreover, this method allows to control the compositional questions created, proving effective in creating a cheap and scalable dataset.

**2.6 Adding distractors to the context** *Images.* Questions from the IMAGELISTQ operator require reasoning over a list of images from the same column, and hence do not require additional distractors. For IMAGEQ questions (single-image), we randomly add images that are associated with the *WikiEntities* that appear in the table, setting a maximum of 15 distractors per question.

*Text.* We used DPR (Karpukhin et al., 2020), a neural information retrieval model, to retrieve distractors for all questions. Each context includes exactly 10 paragraphs, where 1-2 are gold paragraphs and the rest are distractors. Specifically, we encode the first 2 paragraphs of each Wikipedia article with the DPR encoder, and use as distractors the paragraphs with the highest dot product between their encoding and the question encoding. We do not allow: (a) an overlap between the distractors in the training and evaluation sets, (b) distractors originating from the gold article, (c) distractors containing an exact match to the gold answer.

To summarize, each of our examples contains a question, an answer, the formal representation of the PL question (ignored by our models), and all distractors and gold context for all modalities. This renders *MMQA* useful for both open-domain multimodal QA, as well as context-dependant QA.

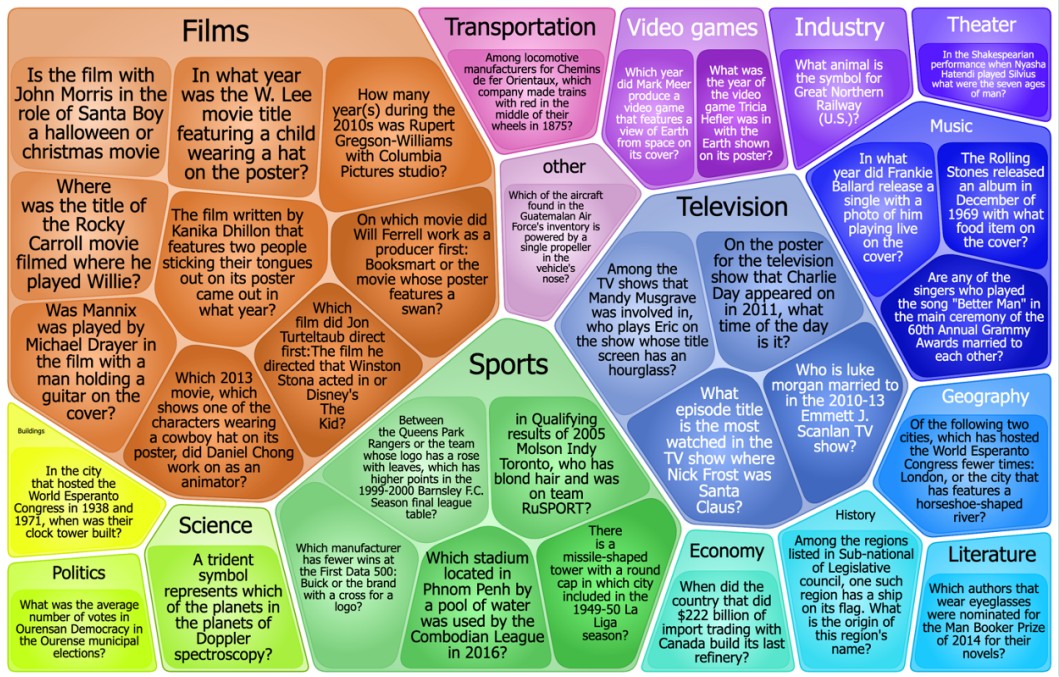

Figure 3: Domain diversity in *MMQA*. The area of each color corresponds to the topic frequency in the dataset.

## 3   DATASET ANALYSIS

To highlight the diversity of *MMQA* we analyze its key statistics, domains, and lexical richness.

**Key Statistics** *MMQA* contains $29,918$ questions, and their main statistics are in Table 3. Since we focus on multimodality, we upsample the number of multimodal questions in the dev. and test sets compared to the training set. Also, about $60\%$ of the questions in *MMQA* are compositional. Questions are relatively long ($18.2$ words), but answers tend to be short ($2.1$ words). The answer for each question can be a single answer or a list of answers. While list answers comprise only $7.4\%$ of the data, when considering compositional questions that contain an intermediate question within them, the proportion of list answers in intermediate questions is higher ($18.9\%$).

| Measurement | Value |
|---|---|
| # Distinct Questions | 29,918 |
| Train multimodal questions | 34.6% |
| Dev.+test multimodal questions | 40.1% |
| Train compositional questions | 58.8% |
| Dev.+test compositional questions | 62.3% |
| Average question length (words) | 18.2 |
| Average # of answers per question | 1.16 |
| List answers | 7.4% |
| List answers per intermediate question | 18.9% |
| Average answer length (words) | 2.1 |
| # of distinct words in questions | 49,649 |
| # of distinct words in answers | 20,820 |
| # of distinct context tables | 11,022 |

Table 3: Key statistics for *MMQA*.

**Domain Diversity** Figure 3 shows a sample of questions from *MMQA* categorized to different domains. While entertainment categories occupy a large portion of our dataset (Films 36%, TV 19%), we observe questions represent a wide variety of topics.

**Lexical Richness** Workers received a bonus when substantially modifying the PL questions. We observe that the average normalized edit distance between the NL questions and the PL questions is high ($0.7$), that NL questions are shorter (avg. length of $20.02$ vs. $22.16$ words for PL questions), and use a richer vocabulary (#unique words $39,319$ vs $37,108$).

## 4   MODELS

Here we present our baseline models. We first train models that interact with a single modality given a question (§4.1), and use those as building blocks in our multimodal approaches (§4.2). We denote the question by $Q$, context paragraphs by $\mathcal{P}$, Table by $T$ and context images by $\mathcal{I}$.

## 4.1 SINGLE-MODALITY QA MODULES

**Text QA Module** Following prior work (Min et al., 2019a; Asai et al., 2020), our text QA module takes as input a question $Q$ and a paragraph $p \in \mathcal{P}$ and answers $Q$ by selecting a span in each paragraph $p$ independently, predicting the start and end positions (Devlin et al., 2019). Additionally, the model returns four scores for for every paragraph $p$ corresponding to: if the answer is (*i*) a span in $p$; (*ii*) *"yes"*; (*iii*) *"no"*; or (*iv*) not in $p$. At inference time, the model selects the paragraph that has the lowest score for *iv* – the answer is not the paragraph. Our model is based on a pre-trained RoBERTa-large model (Liu et al., 2019), fine-tuned on *MMQA*.

**Table QA Module** Following prior work (Herzig et al., 2020), our table QA module takes as input the question $Q$ and the table $T$, and selects a subset of the table cells and an aggregation operation to compute the final answer. Specifically, we linearize the table $T$ by rows, with column names prepended to the corresponding cells (Chen et al., 2019). For example, this converts the table in Figure 1 to the following text: *"Row 1: year is 1957; title is a dangerous age; role is David. Row 2..."*. Next, we concatenate the question to the linearized table, and encode them using RoBERTa-large. We then pass the contextualized representation of every token in the table cell to a linear classifier that computes the probability of the token being selected. The score for a cell is the average of its tokens. Cells with probability $> 0.5$ are selected. Finally, another linear multi-class classifier predicts an aggregation operation from SUM, MEAN, COUNT, YES, NO, and NONE. Aggregation operations are applied on the selected cells, YES and NO operations output *"yes"* or *"no"*, and the NONE operation outputs all selected cells.

**Image QA Module** Questions with visual information are handled by a multimodal transformer that processes the text question and pre-computed image features. For a question $Q$ and a set of images $\mathcal{I}$, we feed the model the question and the visual features $\Phi(i)$ extracted from each image $i \in \mathcal{I}$, along with the name of the *WikiEntity* associated with the image. For each image and question, the model predicts an answer from a fixed vocabulary determined by the answers in the training set and 3 special tokens: $y_{\text{dtr}}$, $y_{\text{p}}$ and $y_{\text{n}}$. In questions where the expected answer is a phrase (e.g., ImageQ), we return the answer from the image where $p(y_{\text{dtr}})$ is lowest (similar to text QA). In questions where the expected answer is a subset of the images (e.g., Compose(TableQ,ImageQ)), we return all images where $p(y_{\text{p}}) > p(y_{\text{n}})$. Our model is based on the pre-trained model VILBERT-MT[1] (Lu et al., 2020). Visual features are extracted by a vision network $\Phi$, comprised of a Faster R-CNN (Ren et al., 2015) pre-trained on Visual Genome (Krishna et al., 2017).

## 4.2 MULTIMODALITY QA MODELS

We turn to models that interact with multiple modalities.

**Multi-Hop Implicit Decomposition (*ImplicitDecomp*)** Our dataset is designed to test reasoning across modalities. As a first attempt towards this goal, we introduce a 2-hop implicit decomposition baseline, capable of combining information scattered across modalities (illustrated in Figure 4).

We first train a question-type classifier, based on RoBERTa-large, that takes a question $Q$ as input, and predicts one of the 16 possible question types (Table 2). The question type can be viewed as a program, specifying the relevant modalities, their order, and the logical operations. For example, if the question type is *Compose(TextQ,TableQ)*, the first hop should be conducted on the table $T$, and the second hop on the paragraphs $\mathcal{P}$. In each hop, we feed the model with the question $Q$, the question type, the hop number, and the context of the corresponding modality. The model automatically identifies which part of the question is relevant at the current hop and does not explicitly decompose the question into sub-questions (hence the name *implicit decomposition*). In the second hop, answers from the first hop are also given as input so that the model can leverage this information and conduct cross-modal reasoning to output the final answer. For all single-modality question types (such as *TextQ* and *TableQ*), the model uses only the first hop to get the answer.

**Single-Hop Routing (*AutoRouting*)** A simple approach for answering questions without cross-modal reasoning is to first determine the modality where the answer is expected to occur, and then run the corresponding single-modality module. We use the aforementioned question type classifier to

---

[1]The multi-task version of VILBERT is used, since it was shown in Lu et al. (2020) that fine-tuning task-specific models from the multi-task model is generally beneficial for performance on single tasks.

determine the modality where the answer will appear, route the question and the context for the predicted modality into the corresponding module, and use the output as the final answer.

**Question-only and Context-only baselines** We run the question-only and context-only baselines, suggested by Kaushik & Lipton (2018). Our question-only baseline is BART-large (Lewis et al., 2019): a sequence-to-sequence model that directly generates the answer given the question. For the context-only baseline, we first predict the question type using the classifier described above to pick a target module. We then feed the relevant context to the target module, replacing the question with an empty string.

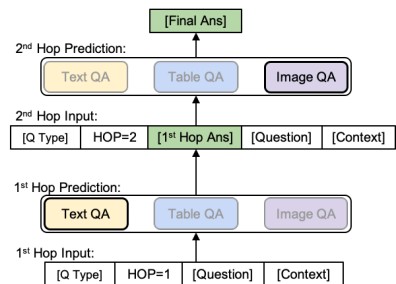

### 4.3 Training and Supervision

Our dataset provides rich supervision including not only the final answer but also question types and intermediate results. Therefore, we can train the pipeline modules in a supervised fashion. Specifically, we train the question type classifier using a cross entropy loss w.r.t the gold question type. For *AutoRouting*, each QA module is trained with the subset of samples whose final answer can be extracted from the corresponding modality. For *ImplicitDecomp*, only one model

Figure 4: *ImplicitDecomp*: Modules with the same color share parameters. In this example, the text QA module is activated to produce the 1st-hop answer, and this intermediate answer is fed into the Image QA model to produce the final answer. Question type ([Q Type]) is determined by a separate classifier.

is trained per modality, which is used to answer both the first-hop and second-hop questions. The question-only and context-only baselines are trained in the corresponding format.

## 5 Experiments

We evaluate models in three different setups: (1) questions that require a single modality to answer (*Single Modality*); (2) questions that require reasoning over multiple modalities (*Multi Modality*); (3) and all questions (*All*). Our evaluation metrics need to support lists of answers, and thus we use average $F_1$ and Exact Match (EM), as described in Dua et al. (2019), where answers on the gold and predicted lists are aligned. Human performance is estimated with 9 expert annotators, who answered 145 questions. Test results are are reported using a single run (one random seed).

We show results in Table 4.[2] *ImplicitDecomp* achieves significantly higher performance (55.9 $F_1$) compared to the other baselines, but lower than human performance (91.2 $F_1$ with provided context, and 84.8 $F_1$ in the open-domain setting over all of Wikipedia), suggesting ample room for improvement. On the *Multi Modality* subset, *ImplicitDecomp*

|  | Single Modality | | Mutli Modality | | All | |
|---|---|---|---|---|---|---|
|  | EM | $F_1$ | EM | $F_1$ | EM | $F_1$ |
| Question-only[2] | 14.2 | 17.0 | 16.9 | 19.5 | 15.3 | 18.0 |
| Context-only | 8.0 | 10.2 | 6.6 | 8.5 | 7.4 | 9.5 |
| *AutoRouting* | 48.9 | 57.1 | 32.0 | 38.2 | 42.1 | 49.5 |
| *ImplicitDecomp* | **51.1** | **58.8** | **46.5** | **51.7** | **49.3** | **55.9** |
| Human | 87.9 | 92.5 | 84.8 | 90.1 | 86.2 | 91.2 |

Table 4: Test set results

substantially improves performance compared to *AutoRouting* (38.2 → 51.7), emphasizing the superiority of our approach on multi-hop questions, while on single-hop questions this gap is smaller.

Since automatic evaluation of performance is non-trivial in our setup, we also manually evaluate human performance. In 94.5% of the cases, answers are either identical or semantically equivalent to the gold answer, 0.7% have an error in the question, and 4.8% are human errors. Human errors are owing to the length of the context, resulting in human fatigue (which models do not suffer from).

**Analysis** To demonstrate that *ImplicitDecomp* indeed performs multi-hop reasoning, successfully answering intermediate questions, we analyze *ImplicitDecomp* predictions for multimodal questions generated using the *Compose*, *Compare* and *Intersect* operations (Table 5). For these questions, we find that when the 1st-hop answer is correct, the model achieves an $F_1$ of 63.9, whereas when the 1st-hop prediction is incorrect, the $F_1$ drops to 37.4. This suggests that the model relies on the 1st-hop

---

[2]We conjecture that the higher performance exhibited by the question-only baseline compared to the context-only baseline is due to the fact that in comparison questions the model needs to choose one answer from two candidates, of which one usually appears in the question, allowing the model to obtain 50% accuracy by guessing.

| Type | Question | 1st-hop prediction | Final prediction | 1st-hop $F_1$ | $F_1$ |
|---|---|---|---|---|---|
| Compose | *What part did Kym Karath play in the TV show whose poster features a dog?* | Lassie | Kathy Vaughn | 62.3 | 50.8 |
| Compare | *Which video game was Wes Johnson involved in earlier: Fallout 4 or the game whose cover shows a gun-wielding man?* | Hammer & Sickle | Hammer & Sickle | 55.7 | 61.1 |
| Intersect | *Which album, released in December of 2011, has a man wearing sunglasses on its cover, and was released under the RCA label?* | TY.O, Back to Love | Back to Love | 33.5 | 55.1 |

Table 5: Examples where *ImplicitDecomp* correctly answers both the intermediate and the entire question, and a breakdown of the 1st-hop $F_1$ and final $F_1$ for the three logical operations: Compose, Compare, and Intersect.

answer, effectively performing multi-hop reasoning. Last, our question type classifier obtains a high accuracy of 91.5% on the test set.

To test whether the compositional questions created are indeed multi-hop, we conducted a qualitative analysis over 50 questions as suggested by (Min et al., 2019b). We find that 6% are of the *Weak Distractors* category, that is, questions such as *"What year... "* when there is only one year appearing in the context, making the question easy. 2% have *Redundant evidence*, that is, questions such as *"Which Donald Trump TV show has ..."* where there is only one TV show starring Donald Trump, making the rest of the question redundant. The remaining 92% indeed require multi-hop reasoning.

# 6 RELATED WORK

Visual question answering—i.e., the task of answering questions about images—has been widely explored in previous work (Antol et al., 2015; Zhang et al., 2016; Goyal et al., 2017; Johnson et al., 2017; Hudson & Manning, 2019; Zellers et al., 2019; Singh et al., 2019; Methani et al., 2020), ranging from synthetic images to scientific plots. Our work differs significantly from those, by including more complex, multi-hop questions that require reasoning over text, tables and images. Currently, the most successful paradigm in VQA is fine-tuning models pre-trained on large amounts of image captioning data (Tan & Bansal, 2019; Lu et al., 2019; 2020; Su et al., 2020; Chen et al., 2020c; Li et al., 2020), an approach we follow for answering image-related questions.

MANYMODALQA (Hannan et al., 2020) move beyond directing the question to an image-only context, to choosing between an image, a text, and a table. Their work focuses on routing the question to the correct context modality. Our question-type classifier, based on RoBERTa-large reaches an accuracy of 91.4% on our 16 possible question types, showing that the main challenge in MMQA is reasoning over the context rather than identifying the question type.

HYBRIDQA (Chen et al., 2020b) presents a cross-modality reasoning challenge over tabular and textual data. A fundamental difference is that our setup offers cross-modality reasoning over images as well. In addition, our approach is cheaper to annotate since it requires only paraphrasing, and the question type distribution is more controllable (we offer 16 major question types vs. 6 in HYBRIDQA). Moreover, our text passages are chosen using the question, answer and table, while in HYBRIDQA only WikiEntities from the table are used to find text passages.

The model proposed in HYBRIDQA introduces a heuristic for linking the text passage to the table cells, which may lead to performance degradation. Conversely, our model uses (automatically-annotated) intermediate multi-hop answers, to perform reasoning and linking implicitly over the full table and text, which should lead to more robust reasoning, in particular when reasoning over multiple table cells, as well as for narrative tracking and co-reference over the full text. In parallel to this work, a new open-domain variant of HYBRIDQA has been released by Chen et al. (2020a).

# 7 CONCLUSION

We present *MMQA*, a new QA dataset that contains 29,918 examples, 35.7% of which require cross-modality reasoning. We describe a novel framework for generating complex multimodal questions at scale, and showcase the diversity and multimodal properties of the resulting dataset. We evaluate *MMQA* using a variety of models, and confirm that the best model exploits the multimodality of the dataset and takes into account multi-hop reasoning via implicit decomposition. However, human performance substantially exceeds the best model, establishing the need for further research involving multiple modalities in question answering systems, which we hope that our work will drive.

## 8 ACKNOWLEDGMENTS

We thank our colleagues at The Allen Institute of AI, James Ferguson and Amir Globerson. This research was partially supported by The Blavatnik Computer Science Research Fund and The Yandex Initiative for Machine Learning, and the European Union's Seventh Framework Programme (FP7) under grant agreement no. 802800-DELPHI. Special thanks to Carrot Search which allowed use to use their FoamTree visualization.

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

## A APPENDIX

Please see separate file for supplementary material.

