# OpenReview forum: "MultiModalQA: complex question answering over text, tables and images"
_ICLR.cc/2021/Conference — ICLR 2021 Poster_

### Official Review · AnonReviewer1 · 2020-10-29
**I think this dataset is new but the necessary of this dataset to QA community is not clear**

**Rating:** 6
**Confidence:** 1

**Review:**

This paper builds a new large-scale QA dataset for multiple modality reasoning including table, text and images.

There are several related works: 1) HYBRIDQA, a dataset that requires reasoning over tabular and textual data. The difference between the proposed dataset and this one is that HYBRIDQA does not require visual inference. 2) MANYMODALQA: a dataset where the context for each question includes information from multiple modalities. The difference between the proposed dataset and this one is that
The answer to each question in MANYMODALQA can be derived from a single modality. Thus, the main difference from others is that the proposed method adds the table information into the framework while the necessary of table for QA task is not clear in this paper. I think almost all our cases of QA have not have the table in our daily life.

The methodology for creating MMQA includes three steps: Context construction, Question generation and Paraphrasing.

This paper also introduces one model to deal with the problem of multiple modality reasoning defined by this new dataset. It seems that these models are not new, which are deeply related to several prior works.

Overall, I think this dataset is new but the necessary of this dataset to QA community is not clear. Moreover, since I am not expert in this area, I am not sure the contribution of this dataset meets the standard of ICLR.

---

> ### Author Response · Authors · 2020-11-23
> **Response to AnonReviewer1**
>
> Thank you for your review. As stated in the second paragraph of the introduction, our dataset MultiModalQA is different from ManyModalQA, since it focuses on questions that require reasoning over multiple modalities for a single question. It is also different from HybridQA in many details, but importantly it incorporates the visual domain. We argue these are important distinctions that will be useful for the research community.
>
> While our model is inspired by prior work, we are unaware of prior papers that perform implicit decomposition as described in the paper.

---

### Official Review · AnonReviewer2 · 2020-10-31

**Rating:** 8
**Confidence:** 3

**Review:**

This paper presents a question answering dataset that needs up-to reasoning over three distinct modalities — text, wikipedia tables, and images of wikipedia entities. The dataset generation step consists of starting with a wiki table, finding entities in the table, followed by finding images associated with the entities and text from existing reading comprehension datasets such as Natural Questions, HotpotQA etc. Next, they generate single-modality question that can be answered from each mode. Additionally the paper also introduces a grammar for generating, compositional questions from the single-mode questions that needs reasoning across modalities, which I believe would be widely useful for creating future datasets. The grammar lets them scale fairly easily. Lastly, the questions generated from the grammar are then paraphrased by annotators. Incentives were given to workers to produce diverse questions (such as bonus if the second-paraphrase of a question was not answered by a baseline model). The dataset comes in both an open-setting as well as closed setting, in which distractors are chosen carefully (e.g. distractor images for other entities in the table or using a state-of-the-art retriever to get paragraphs).

The dataset is tested on a model that uses a pretrained model for each modality. A classifier is used to predict the type of question (which is easy to do) and then each model is applied following the grammar. Following cautionary related work which show that models often take advantage of unwanted bias in the data, they also have a context-only and question-only baseline. They also did a manual analysis revealing that 86% of the questions actually need strong multi-hop reasoning. There is significant gap as expected between human performance and the correct model

Strengths

1. I think this dataset would be a useful test-bed for multi-modal models and several models will be build around it.
2. The grammar used for generating compositional question will also be helpful for building future datasets, so I think that is also an important contribution
3. It looks like the authors have taken sufficient caution to weed out unwanted biases from the dataset
4. The paper is very clearly written. The analysis were also helpful.

Weakness:
1. As rightfully acknowledged in the paper, the distribution of questions is quite different from the question human (currently) ask to a system. However, ability to reason over multiple modalities is important and this dataset is important wrt that goal.

---

> ### Author Response · Authors · 2020-11-23
> **Response to AnonReviewer2**
>
> Thank you for the comments! We address one of the concerns below:
>
> > “As rightfully acknowledged in the paper, the distribution of questions is quite different from the question human (currently) ask to a system. However, the ability to reason over multiple modalities is important and this dataset is important wrt that goal.”
>
> We agree with this characterization. Generated questions are not sampled from a natural distribution, but are useful for work on reasoning over multiple modalities.

---

### Official Review · AnonReviewer4 · 2020-11-02
**A new dataset, some concerns about previous related work and model**

**Rating:** 6
**Confidence:** 3

**Review:**

The paper introduces MultiModalQA, a dataset that requires joint reasoning over table, text and images. The dataset has been created in a semi-automatic way through Wikipedia tables, the Wikientities in them, their related images and related textual question answer pairs from known text QA datasets. For the collection of the dataset, authors collect single modality questions and then use a programmatic way to generate the multimodality versions. The paper also introduces a multi-hop baseline that guesses the question type and then does two hops over the different single modality modules to generate the final answer.

The dataset would be a useful resource for multimodal QA advancement but the manuscript in the current form disregards all of the prior work that has happened in the space including the work on VQA [1], TextVQA [2] and PlotQA[3]. There has been quite a lot of work on multimodality QA and the paper should address those and take learnings from those papers.
Specifically, the model introduced in the paper is too crude and specific to the dataset generation scheme (like depending on the question type) which limits the applicability of the dataset as well as the further approaches that will be developed on this dataset. For example, M4C [4] a model on TextVQA uses transformers by projecting different modalities into the same space and concatenating them as a single sequence. Comparison against similar approaches on MultiModalQA would be more useful compared to hardcoded approaches like ImplicitDecomp. This approach reminds me of the approaches that were developed on CLEVR using program synthesis and weren’t useful in the long run. Similar to TableQA modules, tables + text + images can be input to the M4C transformer as a single sequence and the classification accuracy can be calculated.

Some other questions to the authors:
- Is there any specific reason to use ViLBERT-MT instead of ViLBERT as ViLBERT finetuned on your task would be more powerful - compared to using ViLBERT-MT.

- How many runs were done to compute the metrics in Table 3.

- Have you done other ablations on 2-hop setting by using the TextQA module twice, TableQA module twice etc. It would allow us to understand better which module matters most.

Overall, this is an interesting and useful dataset. I would recommend acceptance if the concerns and writing about previous works on multimodality are fixed and my other questions and concerns are considered.


[1] Goyal, Yash, et al. "Making the V in VQA matter: Elevating the role of image understanding in Visual Question Answering." Proceedings of the IEEE Conference on Computer Vision and Pattern Recognition. 2017.

[2] Singh, Amanpreet, et al. "Towards vqa models that can read." Proceedings of the IEEE Conference on Computer Vision and Pattern Recognition. 2019.

[3] Methani, Nitesh, et al. "PlotQA: Reasoning over Scientific Plots." The IEEE Winter Conference on Applications of Computer Vision. 2020.

[4] Hu, Ronghang, et al. "Iterative answer prediction with pointer-augmented multimodal transformers for textvqa." Proceedings of the IEEE/CVF Conference on Computer Vision and Pattern Recognition. 2020


Edit after rebuttal: I have read the author response and thank the authors for their comments and answers to my questions. I would like to keep my rating as it is. I recommend authors to tone down the claims around being first MultimodalQA dataset and position themself properly with respect to previous related work if accepted.

---

> ### Author Response · Authors · 2020-11-23
> **Response to AnonReviewer4**
>
> Thank you for the comments!
>
> We will definitely elaborate on previous works in multimodality when we have one more page in camera-ready. Unfortunately, we left more detailed comparisons out on this version, due to space limitations.
>
> We address some of the concerns and questions below:
>
> >“Specifically, the model introduced in the paper is too crude and specific to the dataset generation scheme (like depending on the question type) which limits the applicability of the dataset as well as the further approaches that will be developed on this dataset. “
>
> Thank you for your comment. We note that our model, which is meant to be a starting point for work on this problem, simply supports composition, conjunction and comparison, which are general-purpose operations that occur in many past datasets for reasoning (HotpotQA, HybridQA, inter alia.). Thus, we argue that our model constitutes a reasonable baseline for such scenarios where answering questions requires reasoning over these operations.
>
> >“This approach reminds me of the approaches that were developed on CLEVR using program synthesis and weren’t useful in the long run. Similar to TableQA modules, tables + text + images can be input to the M4C transformer as a single sequence and the classification accuracy can be calculated.”
>
> We agree that other approaches are possible, but there has been ample work on models that are interpretable and expose their reasoning explicitly [1, 2, 3]. We note that since the context size is long, it is non-trivial to have a single transformer process all of the inputs.
>
> [1] Chen, Xinyun, et al. "Neural symbolic reader: Scalable integration of distributed and symbolic representations for reading comprehension." International Conference on Learning Representations. 2019.
>
> [2] Herzig, Jonathan, et al. "TAPAS: Weakly Supervised Table Parsing via Pre-training." arXiv preprint arXiv:2004.02349 (2020).
>
> [3] Yin, Pengcheng, et al. "TaBERT: Pretraining for Joint Understanding of Textual and Tabular Data." arXiv preprint arXiv:2005.08314 (2020).
>
>
> >“Is there any specific reason to use ViLBERT-MT instead of ViLBERT as ViLBERT finetuned on your task would be more powerful - compared to using ViLBERT-MT.”
>
> Thanks for the question. The authors of VILBERT-MT demonstrate in [1] that fine-tuning task-specific models from the multi-task model is generally beneficial to performance at single tasks.
>
> [1] Lu, Jiasen, et al. "12-in-1: Multi-task vision and language representation learning." Proceedings of the IEEE/CVF Conference on Computer Vision and Pattern Recognition. 2020.
>
> >“How many runs were done to compute the metrics in Table 3.”
>
> If we understand the question correctly, the test results are from a single run. Additionally, we conducted multiple runs for hyperparameter search when training each single modality module, especially for tuning the learning rate and the number of training epochs. The test results in Table 3 were obtained by using the best checkpoint of each single modality module.
>
> >“Have you done other ablations on 2-hop setting by using the TextQA module twice, TableQA module twice etc. It would allow us to understand better which module matters most."
>
> Thanks for the suggestion. We conducted this ablation by evaluating the same module twice on all questions that require reasoning over multiple modalities (Multi-Modality) in the dev set. Below are the EM scores for each modality the final answer lays, compared to our original ImplicitDecomp results:
>
> |                          | Final answer from text | Final answer from table | Final answer from image |
> |--------------------------|------------------------|-------------------------|-------------------------|
> | TextQA \+ TextQA         | 33\.1                  | 7                       | 7\.4                    |
> | TableQA \+ TableQA       | 1\.6                   | 26\.9                   | 0                       |
> | ImageQA \+ ImageQA       | 0                      | 5\.9                    | 24\.3                   |
> | Our ImplicitDecomp Model | 47\.3                  | 50\.4                   | 33\.1                   |
>
> Results show a sharp drop in accuracy (47.2 → 33.1 in text, 50.4 → 26.9 in tables and 33.1 → 24.3 in images), suggesting that the model is heavily relying on the intermediate answer, and that correctly answering the questions requires 2-hops.

---

### Official Review · AnonReviewer5 · 2020-11-04

**Rating:** 6
**Confidence:** 3

**Review:**

### Summary
The authors present a new dataset, MultiModalQA, with the intent of measuring a model’s ability to reason across different modalities (free text, structured tables, and images) in question answering, and in which a large percentage of the questions requires cross-modal reasoning. The authors provide a detailed look at the framework used to generate questions in the context of several modalities. Finally, the authors propose a multimodal model that performs multi-hop reasoning (removing the need for explicit question decomposition), which outperforms strong baselines, but is still far behind human performance, indicating that the task is nontrivial and would benefit the research community.

### Positives
- The dataset is interesting and comprehensive, covering multiple modalities with a significant portion covering compositional questions as well.
- The authors describe in detail their method of dataset construction, leveraging existing datasets in the literature and providing insight for others should they so desire to construct their own multimodal qa dataset.
- In the process, the authors consider and justify shortcomings of the dataset (e.g., that the automatic generation of examples may be seen as limiting the question distribution to a non “natural” source)
- The authors consider a baseline multi-hop reasoning model that highlights the need for cross-modal reasoning to achieve good performance on the dataset.

### Negatives
- The paper could benefit from further comparison and exploration of how the methods and data relates to others in the literature.
- I would have liked to have seen a little bit more comparison between this dataset and others in terms of raw statistics - the authors mention HybridQA and ManyModalQA in the introduction, so comparing to those would be a start.
- Additionally, the authors could have provided some analysis of their proposed model’s effectiveness on other datasets - does the model still work well for single-modality QA datasets when compared to other pre-existing models? Does the “question classifier” generalize to questions not within the 16-construction PL framework?

### Decision
I think this paper is marginally above the acceptance threshold. The dataset is unique and well-constructed, and the multimodal/QA community would benefit from its use. The authors consider solid baseline models for the task, though the paper would have benefitted from having more exploration of how well these models generalize to other tasks, and additionally how this task compares to other similar ones in the domain.

### Questions
Please address negatives; in addition:

1. There are multiple image-based QA datasets out there (e.g. VQA) with several image-based questions already - did you consider automatic matching of images from WikiEntities to these datasets? As it would eliminate the need for crowdsourcing the image single-modality questions.
2. The authors mentioned a feedback mechanism for AMT workers such that they were given bonuses if a baseline model answered the question correctly before & after rephrasing; at first glance this seems to have introduced some bias in that it makes the questions easier for models to answer. Was this considered when adding this feedback mechanism?

---

> ### Author Response · Authors · 2020-11-23
> **Response to AnonReviewer5**
>
> Thank you for the comments! We address some of the concerns and questions below:
>
> >“The paper could benefit from further comparison and exploration of how the methods and data relates to others in the literature.
> I would have liked to have seen a little bit more comparison between this dataset and others in terms of raw statistics - the authors mention HybridQA and ManyModalQA in the introduction, so comparing to those would be a start.”
>
> Thank you for this suggestion. We will include this in the final version when we have one more page, unfortunately, we left these statistics out on this version, due to space limitations.
>
> >“Additionally, the authors could have provided some analysis of their proposed model’s effectiveness on other datasets - does the model still work well for single-modality QA datasets when compared to other pre-existing models? Does the “question classifier” generalize to questions not within the 16-construction PL framework?.”
>
> Thanks for this idea. We tested zero-shot question type classification performance of our BERT-based modality classification model on ManyModalQA, a collection of single-hop questions across three modalities.
> As our original question classification model is a 16-way classification, we mapped the 16 types into 3 types based on the modality from which the final answer would be extracted (e.g., Compose(text, table) --> text).
> The zero-shot accuracy on ManyModalQA is 65.39 %., whereas the original paper’s classification results after fine-tuning is  77.93%. For the final version, we will also fine-tune on ManyModalQA and compare our classifier accuracy.
>
> >“There are multiple image-based QA datasets out there (e.g. VQA) with several image-based questions already - did you consider automatic matching of images from WikiEntities to these datasets? As it would eliminate the need for crowdsourcing the image single-modality questions.”
>
> Thanks for the question. To compose image questions with other modalities we require questions about wikipedia entities, annotating questions on wikipedia pictures enables us to identify which wikipedia entity is related to the picture. We also require wikipedia images to have good distractors, as well as ask about the list of entities present in the wikipedia table. None of the datasets above annotate questions specifically from wikipedia, thus we needed to annotate new questions ourselves.
>
> >“The authors mentioned a feedback mechanism for AMT workers such that they were given bonuses if a baseline model answered the question correctly before & after rephrasing; at first glance this seems to have introduced some bias in that it makes the questions easier for models to answer. Was this considered when adding this feedback mechanism?”
>
> Thanks for the question. As we note in section 2.5:  “workers received a bonus if a baseline model correctly answered the question after their first paraphrasing attempt, but incorrectly after they refined the paraphrase”, we used the last as the final rephrased question. We observed larger variation in language while preserving the semantics of the artificial question.

---

### Decision · Program_Chairs · 2021-01-07
**Final Decision**

**Decision:**

Accept (Poster)

**Comment:**

The paper presents a new dataset for multimodal QA that is deemed interesting, relevant and well executed by all reviewers. Multimodality in NLP (QA included) is an increasingly important topic and this paper provides a potentially impactful benchmark for research in it. All reviewers acknowledge that.

We hence recommend to accept this paper as a poster. We recommend the authors to further improve the draft before camera ready by using the recommendations made by the reviewers with a particular focus on an extended discussion wrt prior work on VQA and other. The paper should also add more precisions on the license(s) related to the images used in the dataset.